

# Four-body Faddeev-type calculation of the $\bar{K}NNN$ system: Preliminary results

**Nina V. Shevchenko**⋆

Nuclear Physics Institute of the CAS, Řež 250 68, Czech Republic

⋆ shevchenko@ujf.cas.cz

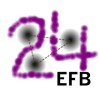

*Proceedings for the 24th edition of European Few Body Conference,
Surrey, UK, 2-6 September 2019*

## Abstract

**The paper is devoted to the $\bar{K}NNN$ system, consisting of an antikaon and three nucleons. Four-body Faddeev-type AGS equations are being solved in order to find possible quasi-bound state in the system.**

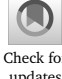
## 1 Introduction

Attractive nature of $\bar{K}N$ integration lead to suggestions, that quasi-bound states can exist in few-body systems consisting of antikaons and nucleons [1]. In particular, a deep and relatively narrow quasi-bound state was predicted in the lightest three-body $\bar{K}NN$ system [2]. Many theoretical calculations of the system were performed after that using different methods and inputs, see e.g. [3]. All of them agree, that the quasi-bound state really exists in spin-zero state of $\bar{K}NN$, usually denoted as $K^-pp$, but predict quite different binding energies and widths of the state.

In recent years we performed a series of calculations of different properties and states of the three-body $\bar{K}NN$ and $\bar{K}\bar{K}N$ systems [3], using dynamically exact Faddeev-type equations in AGS form with coupled $\bar{K}NN$ and $\pi\Sigma N$ channels. In particular, we predicted $K^-pp$ quasi-bound state binding energy and width using three different models of $\bar{K}N$ interaction. The same was done for the $\bar{K}\bar{K}N$ system. We also demonstrated, that there is no quasi-bound states, caused by pure strong interactions, in another spin state of $\bar{K}NN$ system, which is $K^-d$. In addition, we calculated the near-threshold amplitudes of $K^-$ elastic scattering on deuteron. Finally, we evaluated 1$s$ level shift in kaonic deuterium, which is an atomic state, caused by presence of the strong $\bar{K}N$ interaction in comparison to the pure Coulomb state.

The next step is study of a four-body $\bar{K}NNN$ system. Some calculations were already performed for it [1,4,5], but more accurate calculations are needed. We use four-body Faddeev-type equations in AGS form [6]. Only these dynamically exact equations in momentum representation can treat energy-dependent $\bar{K}N$ potentials, necessary for the this system, exactly.

## 2  Four-body Faddeev-type AGS equations for the $\bar{K}NNN$ system

Description of the four-body Faddeev-type equations in AGS form [6] should start from the three-body AGS equations [7] since the three-body transition operators, being solution of the three-body equations, enter the four-body ones together with two-body $T$-matrices. If separable potentials of the form $V_\alpha = \lambda_\alpha |g_\alpha\rangle\langle g_\alpha|$ are used, the three-body transitions operators satisfy the three-body Faddeev-type equations in AGS form [7]:

$$X_{\alpha\beta}(z) = Z_{\alpha\beta}(z) + \sum_{\gamma=1}^{3} Z_{\alpha\gamma}(z)\tau_\gamma(z)X_{\gamma\beta}(z), \tag{1}$$

with transition $X_{\alpha\beta}$ and kernel $Z_{\alpha\beta}$ operators are defined as

$$X_{\alpha\beta}(z) = \langle g_\alpha|G_0(z)U_{\alpha\beta}(z)G_0(z)|g_\beta\rangle, \tag{2}$$
$$Z_{\alpha\beta}(z) = (1-\delta_{\alpha\beta})\langle g_\alpha|G_0(z)|g_\beta\rangle. \tag{3}$$

Operator $U_{\alpha\beta}(z)$ in Eq.(2) is a three-body transition operator of the general form, which describes process $\beta + (\alpha\gamma) \to \alpha + (\beta\gamma)$, while $G_0(z)$ in Eqs.(2,3) is the three-body free Green function. Faddeev partition indices $\alpha,\beta = 1,2,3$ simultaneously define a particle ($\alpha$) and the remained pair ($\beta\gamma$). The operator $\tau_\alpha(z)$ in Eq.(1) is an energy-dependent part of a separable two-body $T$-matrix $T_\alpha(z) = |g_\alpha\rangle\tau_\alpha(z)\langle g_\alpha|$, corresponding to the separable potential describing interaction in the ($\beta\gamma$) pair; $|g_\alpha\rangle$ is a form-factor.

The four-body Faddeev-type AGS equations [6], written for separable potentials, have a form

$$\bar{U}_{\alpha\beta}^{\sigma\rho}(z) = (1-\delta_{\sigma\rho})(\bar{G}_0^{-1})_{\alpha\beta}(z) + \sum_{\tau,\gamma,\delta}(1-\delta_{\sigma\tau})\bar{T}_{\alpha\gamma}^{\tau}(z)(\bar{G}_0)_{\gamma\delta}(z)\bar{U}_{\delta\beta}^{\tau\rho}(z), \tag{4}$$

$$\bar{U}_{\alpha\beta}^{\sigma\rho}(z) = \langle g_\alpha|G_0(z)U_{\alpha\beta}^{\sigma\rho}(z)G_0(z)|g_\beta\rangle, \tag{5}$$

$$\bar{T}_{\alpha\beta}^{\tau}(z) = \langle g_\alpha|G_0(z)U_{\alpha\beta}^{\tau}(z)G_0(z)|g_\beta\rangle, \tag{6}$$

$$(\bar{G}_0)_{\alpha\beta}(z) = \delta_{\alpha\beta}\tau_\alpha(z). \tag{7}$$

Here operators $\bar{U}_{\alpha\beta}^{\sigma\rho}$ and $\bar{T}_{\alpha\beta}^{\tau}$ contain four-body $U_{\alpha\beta}^{\sigma\rho}(z)$ and three-body $U_{\alpha\beta}^{\tau}(z)$ transition operators of the general form, correspondingly. The lower indices $\alpha,\beta$ in Eqs.(4,5,6) similarly to the case of the three-body equations (1,2,3) define two-body subsystems of the full system. The upper indices $\tau,\sigma,\rho$ define a partition of the four-body system which can be of $3+1$ or $2+2$ type. In particular, there are two partitions of $3+1$ type: $|\bar{K} + (NNN)\rangle$, $|N + (\bar{K}NN)\rangle$, - and one of the $2+2$ type: $|(\bar{K}N) + (NN)\rangle$, - for the $\bar{K}NNN$ system. The free Green function $G_0(z)$ acts in four-body space.

The four-body system of AGS equations Eq.(4) look similar to the three-body AGS system with arbitrary potentials. If we, as suggested in [8], represent the "effective three-body potentials" $\bar{T}_{\alpha\beta}^{\tau}(z)$ in Eq. (4) in a separable form $\bar{T}_{\alpha\beta}^{\tau}(z) = |\bar{g}_\alpha^\tau\rangle\bar{\tau}_{\alpha\beta}^\tau(z)\langle\bar{g}_\alpha^\tau|$, the four-body equations can be rewritten as [8]

$$\bar{X}_{\alpha\beta}^{\sigma\rho}(z) = \bar{Z}_{\alpha\beta}^{\sigma\rho}(z) + \sum_{\tau,\gamma,\delta}\bar{Z}_{\alpha\gamma}^{\sigma\tau}(z)\bar{\tau}_{\gamma\delta}^{\tau}(z)\bar{X}_{\delta\beta}^{\tau\rho}(z), \tag{8}$$

with new four-body transition $\bar{X}^{\sigma\rho}$ and kernel $\bar{Z}^{\sigma\rho}$ operators

$$\bar{X}_{\alpha\beta}^{\sigma\rho}(z) = \langle\bar{g}_\alpha^\sigma|\bar{G}_0(z)_{\alpha\alpha}\bar{U}_{\alpha\beta}^{\sigma\rho}(z)\bar{G}_0(z)_{\beta\beta}|\bar{g}_\beta^\rho\rangle, \tag{9}$$

$$\bar{Z}_{\alpha\beta}^{\sigma\rho}(z) = (1-\delta_{\sigma\rho})\langle\bar{g}_\alpha^\sigma|\bar{G}_0(z)_{\alpha\beta}|\bar{g}_\beta^\rho\rangle. \tag{10}$$

Necessary for the $\bar{K}NNN$ calculations $\bar{K}N$ and $NN$ potentials, which we use, are separable ones by construction. Therefore, we need to construct separable versions of three-body and 2+2 amplitudes only, entering the equations (8). We use Energy Dependent Pole Expansion/ Approximation (EDPE/ EDPA) method, suggested in [9] specially for the four-body AGS equations.

Three-body Faddeev-type AGS equations written in momentum basis for $s$-wave interactions have a form:

$$X_{\alpha\beta}(p,p';z) = Z_{\alpha\beta}(p,p';z) + \sum_{\gamma=1}^{3} 4\pi \int_0^\infty Z_{\alpha\gamma}(p,p'';z)\tau_\gamma(p'';z)X_{\gamma\beta}(p'',p';z)p''^2 dp'', \quad (11)$$

were $p, p'$ and $z$ are three-body momenta and energy. Eigenvalues $\lambda_n$ and eigenfunctions $g_{n\alpha}(p;z)$ of the system Eq.(11) can be evaluated from the system of equations

$$g_{n\alpha}(p;z) = \frac{1}{\lambda_n} \sum_{\gamma=1}^{3} 4\pi \int_0^\infty Z_{\alpha\gamma}(p,p';z)\tau_\gamma(p';z)g_{n\gamma}(p';z)p'^2 dp', \quad (12)$$

with normalization condition

$$\sum_{\gamma=1}^{3} 4\pi \int_0^\infty g_{n\gamma}(p';z)\tau_\gamma(p';z)g_{n'\gamma}(p';z)p'^2 dp' = -\delta_{nn'}. \quad (13)$$

EDPE/EDPA method needs solution of the eigenequations Eq.(12) for a fixed energy $z$, which usually is chosen to be the binding energy $z = E_B$. After that energy dependent form-factors

$$g_{n\alpha}(p;z) = \frac{1}{\lambda_n} \sum_{\gamma=1}^{3} 4\pi \int_0^\infty Z_{\alpha\gamma}(p,p';z)\tau_\gamma(p';E_B)g_{n\gamma}(p';E_B)p'^2 dp' \quad (14)$$

and propagators

$$(\Theta(z))_{mn}^{-1} = \sum_{\gamma=1}^{3} 4\pi \int_0^\infty g_{m\gamma}(p';z)\tau_\gamma(p';E_B)g_{n\gamma}(p';E_B)p'^2 dp'$$
$$- \sum_{\gamma=1}^{3} 4\pi \int_0^\infty g_{m\gamma}(p';z)\tau_\gamma(p';z)g_{n\gamma}(p';z)p'^2 dp' \quad (15)$$

can be calculated. Finally, the separable version of a three-body amplitude has a form

$$X_{\alpha\beta}(p,p';z) = \sum_{m,n=1}^{\infty} g_{m\alpha}(p;z)\Theta_{mn}(z)g_{n\beta}(p';z). \quad (16)$$

If only one term is taken in the sums in Eq.(16), the Energy Dependent Pole Expansion turns into Energy Dependent Pole Approximation. It is seen, that in contrast to Hilbert-Schmidt expansion, EDPE method needs only one solution of the eigenvalue equations Eq.(12) and calculations of the integrals Eqs.(14,15) after that. According to the authors, the method is accurate already with one term (i.e. EDPA), and it converges faster than Hilbert-Schmidt expansion.

In order to find the quasi-bound state energy the homogeneous system of equations Eq.(8) has to be solved. We started by writing down the system Eq.(8) for 18 channels $\sigma_\alpha$ with $\alpha =$

$NN$ or $\bar{K}N$, considering three nucleons as nonindentical particles:

$$
\begin{aligned}
&1_{NN}: |\bar{K} + (N_1 + N_2 N_3)\rangle, |\bar{K} + (N_2 + N_3 N_1)\rangle, |\bar{K} + (N_3 + N_1 N_2)\rangle, \\
&2_{NN}: |N_1 + (\bar{K} + N_2 N_3)\rangle, |N_2 + (\bar{K} + N_3 N_1)\rangle, |N_3 + (\bar{K} + N_1 N_2)\rangle, \\
&2_{\bar{K}N}: |N_1 + (N_2 + \bar{K} N_3)\rangle, |N_2 + (N_3 + \bar{K} N_1)\rangle, |N_3 + (N_1 + \bar{K} N_2)\rangle, \\
&\qquad |N_1 + (N_3 + \bar{K} N_2)\rangle, |N_2 + (N_1 + \bar{K} N_3)\rangle, |N_3 + (N_2 + \bar{K} N_1)\rangle, \\
&3_{NN}: |(N_2 N_3) + (\bar{K} + N_1)\rangle, |(N_3 N_1) + (\bar{K} + N_2)\rangle, |(N_1 N_2) + (\bar{K} + N_3)\rangle, \\
&3_{\bar{K}N}: |(\bar{K} N_1) + (N_2 + N_3)\rangle, |(\bar{K} N_2) + (N_3 + N_1)\rangle, |(\bar{K} N_3) + (N_1 + N_2)\rangle.
\end{aligned}
\tag{17}
$$

After antisymmetrization, necessary for a system with identical fermions, the system of equations to be solved can be written in a matrix form:

$$
\hat{X} = \hat{Z}\,\hat{\tau}\,\hat{X}.
\tag{18}
$$

Our $\bar{K}N$ and $NN$ potentials, which we use for the $\bar{K}NNN$ system calculations, are isospin- and spin-dependent ones. In addition, our $NN$ interaction model is a two-term potential. At the first step only one separable term was used for the three-body $\bar{K}NN$, $NNN$ and 2+2 $\bar{K}N + NN$ amplitudes in Eq. (16) (EDPA). Keeping all this in mind, matrices $\hat{Z}$ and $\hat{\tau}$, entering the antisymmetrized equations Eq.(18) are matrices $18 \times 18$ containing the kernel operators $\bar{Z}_\alpha^{\sigma\rho}$ and $\bar{\tau}_{\alpha\beta}^\rho$, correspondingly.

## 3 Two-body potentials, $3+1$ and $2+2$ partitions

### 3.1 Two-body potentials

Both $\bar{K}N$ and $NN$ potentials are separable isospin- and spin-dependent ones in $s$-wave. We use three our separable antikaon-nucleon potentials constructed for our three-body calculations of the $\bar{K}NN$ and $\bar{K}\bar{K}N$ systems. They are: two phenomenological potentials with coupled $\bar{K}N - \pi\Sigma$ channels, having one- or two-pole structure of the $\Lambda(1405)$ resonance [12] and a chirally motivated model with coupled $\bar{K}N - \pi\Sigma - \pi\Lambda$ channels and two-pole structure [13]. All three potentials describe low-energy $K^-p$ scattering, namely: elastic $K^-p \to K^-p$ and inelastic $K^-p \to MB$ cross-sections and threshold branching ratios $\gamma, R_c, R_n$. They also reproduce $1s$ level shift of kaonic hydrogen caused by the strong $\bar{K}N$ interaction in comparison to the pure Coulomb level, measured by SIDDHARTA experiment [10]: $\Delta_{1s}^{SIDD} = -283 \pm 36 \pm 6$ eV, $\Gamma_{1s}^{SIDD} = 541 \pm 89 \pm 22$ eV. All the experimental data are described by three our potentials with equally high accuracy. In addition, elastic $\pi\Sigma$ cross-sections with isospin $I_{\pi\Sigma}$ provided by all three potentials have a bump in a region of the $\Lambda(1405)$ resonance (according to PDG [11]: $M_{\Lambda(1405)}^{PDG} = 1405.1_{-1.0}^{+1.3}$ MeV, $\Gamma_{\Lambda(1405)}^{PDG} = 50.5 \pm 2.0$ MeV). The poles corresponding to the $\Lambda(1405)$ resonance are situated at

$$
z_{\Lambda(1405)-1}^{SIDD1} = 1426 - i\,48\,\text{MeV}
\tag{19}
$$

$$
z_{\Lambda(1405)-1}^{SIDD2} = 1414 - i\,58\,\text{MeV}, \quad z_{\Lambda(1405)-2}^{SIDD2} = 1386 - i\,104\,\text{MeV}
\tag{20}
$$

for the phenomenological potentials with one- and two-pole structure, correspondingly [13], and at

$$
z_{\Lambda(1405)-1}^{Chiral} = 1417 - i\,33\,\text{MeV}, \quad z_{\Lambda(1405)-2}^{Chiral} = 1406 - i\,89\,\text{MeV}
\tag{21}
$$

for the chirally motivated potential [14].

The three antikaon-nucleon potentials with coupled $\bar{K}N - \pi\Sigma$ channels were used in three-body AGS equations with coupled $\bar{K}NN - \pi\Sigma N$ three-body channels. By this the channel

coupling was taken into account in a direct way. The four-body AGS equations, which we solve, are too complicated to do the same. Due to this we use the exact optical versions of our $\bar{K}N$ potentials. They have exactly the same elastic part of the potential as the potential with coupled channels, while all in-elasticity is taken into account in an energy-dependent imaginary part of the potential. It was demonstrated in our three-body calculations, that such potentials give very accurate results in comparison with the results obtained with the coupled-channel potentials, see below. Due to this we assume that it is a good approximation for the four-body calculations as well.

We constructed a new version of the two-term separable $NN$ potential. It reproduces Argonne v18 $NN$ phase shifts at low energies up to 500 MeV with change of sign, which means it is repulsive at short distances. It provides the following singlet and triplet $NN$ scattering lengths: $a_s = 16.32$ fm, $a_t = -5.40$ fm, and give the deuteron binding energy $E_{deu} = 2.225$ MeV.

No three-body potentials were used since the four-body Faddeev-type equations are too complicated in their original form with "normal" pair potentials already.

### 3.2  $3+1$ and $2+2$ partitions

We are studying the $\bar{K}NNN$ system with the lowest value of the four-body isospin $I^{(4)} = 0$, which can be denoted as $K^- ppn$. Its total spin $S^{(4)}$ is equal to one half, while the orbital momentum is zero, since all two-body interactions are chosen to be $s$-wave ones. For the $\bar{K}NNN$ system with these quantum numbers the following three-body subsystems contribute:

- $\bar{K}NN$ with isospin $I^{(3)} = 1/2$ and spin $S^{(3)} = 0$ ($K^- pp$) or spin $S^{(3)} = 1$ ($K^- d$).

- $NNN$ with isospin $I^{(3)} = 1/2$ and spin $S^{(3)} = 1/2$ ($^3$H or $^3$He).

The three-body $\bar{K}NN$ system with different quantum numbers was studied in our previous works, in particular, quasi-bound state pole positions in the $K^- pp$ system ($\bar{K}NN$ with isospin $I^{(3)} = 1/2$ and spin $S^{(3)} = 0$) were calculated in [14]. The pole positions calculated with coupled $\bar{K}NN$ and $\pi\Sigma N$ channels and $\bar{K}N - \pi\Sigma$ potentials are:

$$z_{K^- pp}^{\text{SIDD1}} = -53.3 - i\,32.4\,\text{MeV}, \tag{22}$$

$$z_{K^- pp}^{\text{SIDD2}} = -47.4 - i\,24.9\,\text{MeV}, \tag{23}$$

$$z_{K^- pp}^{\text{Chiral}} = -32.2 - i\,24.3\,\text{MeV}, \tag{24}$$

while one-channel calculation of the $\bar{K}NN$ system using exact optical $\bar{K}N(-\pi\Sigma)$ potentials give slightly different results:

$$z_{K^- pp,Opt}^{\text{SIDD1}} = -54.2 - i\,30.5\,\text{MeV}, \tag{25}$$

$$z_{K^- pp,Opt}^{\text{SIDD2}} = -47.4 - i\,23.0\,\text{MeV}, \tag{26}$$

$$z_{K^- pp,Opt}^{\text{Chiral}} = -32.9 - i\,24.4\,\text{MeV}. \tag{27}$$

Comparison of the exact results Eqs.(22–24) and the approximated ones Eqs.(25–27) demonstrates that using of the exact optical $\bar{K}N(-\pi\Sigma)$ potentials instead of the coupled-channel $\bar{K}N - \pi\Sigma$ ones is quite accurate approximation.

No quasi-bound states caused by pure strong interactions were found in the $K^- d$ system ($\bar{K}NN$ with isospin $I^{(3)} = 1/2$ and spin $S^{(3)} = 1$). The codes for numerical solution of the three-body AGS equations for the $\bar{K}NN$ systems were modified to construct separable versions of the three-body amplitudes, as described before.

The three-body AGS equations Eq.(1) were written and numerically solved for the three-nucleon system $NNN$ with our new two-term $NN$ potential as an input. The calculated binding energy was found to be 9.95 MeV for both $^3$H and $^3$He nuclei since Coulomb interaction was not taken into account. The numerical code was afterwards changed for construction of separable version of the $NNN$ subsystem.

Finally, the partition of the $2+2$ type $\bar{K}N + NN$ is a system with two non-interacting pairs of particles. It was described by special three-body system of AGS equations, and its separable version was constructed in similar way as in $\bar{K}NN$ and $NNN$ three-body subsystems.

## 4  Preliminary results and conclusion

As the first step we solved the four-body AGS equations for the $K^- ppn$ system, using only one separable term in Eq.(16), which is EDPA method. We used the exact optical versions of two our phenomenological $\bar{K}N$ potentials with one- and two-pole structure of the $\Lambda(1405)$ resonance (our chirally motivated model of the antikaon-nucleon interaction has isospin-dependent form-factors, so that some changes in the four-body program code are necessary). Very preliminary results for the pole position of the quasi-bound state in the $K^- ppn$ are:

$$z^{\text{SIDD1}}_{K^- ppn, Prelim} = -75 - i\,14\,\text{MeV} \tag{28}$$

$$z^{\text{SIDD2}}_{K^- ppn, Prelim} = -71 - i\,8\,\text{MeV}. \tag{29}$$

As in the case of the $K^- pp$ system, the quasi-bound state calculated using one-pole phenomenological $\bar{K}N$ potential (SIDD1) is deeper then that one calculated with the two-pole $V_{\bar{K}N}$, and it has larger width. The results differ from those shown at the conference since some bugs were found in the program code after the talk.

To conclude, the four-body Faddeev-type AGS equations for search of the quasi-bound state in the $\bar{K}NNN$ system were written down. The code for numerical four-body calculations was written, preliminary results for the pole positions were obtained. Tests and checks of the program code are still in progress.

## Acknowledgments

The author is thankful to the Academy of Sciences of the Czech Republic for provided subsidy for DSc. researchers. The work was supported by GACR grant 19-19640S.

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
