# Peer review of "Four-Body Faddeev-Type Calculation of the $\bar{K}NNN$ System: Preliminary Results"

_SciPost Physics Proceedings, doi:SciPost Phys. Proc. 3, 041 (2020)_

## Round 1 · Referee Report · Anonymous · 2019-12-9

Report

SciPost Physics Proceedings

``Four-Body Faddeev-Type Calculation of the K̄ N N N System:
Preliminary Results''

by
N. V. Shevchenko

Studying the binding energies of mesonic nuclei is an important and active
field of research. In that, few-body systems where accurate numerical calculations can be carried out, play a special role. In the current manuscript the author reports on the development of a new 4-body code solving the AGS
equations for the KbarNNN system.

The paper is rather technical, and present two model calculations predicting a very deep 4-body bound state. After some modifications I believe that it can be published on SciPost Physics.

First of all, in Sec. 4 the author writes ``very preliminary results for ...''
and further down ``The results differ from those shown at the conference''.
These remark convey no confidence in the reported results. I advice the author
to either remove these remarks is she trust her results or remove Sec. 4 all-together, and report the results when she gain full confidence in her code.

On top of this, here few more minor comments:
- In the introduction the author writes ``more accurate calculations are needed'', but no argument is given. Does she suspects there is an error in the
reported results? does she has a better input?
- ``separabelize'' is not a word. Same for its derivatives.
- The second term in Eq. (15) is just a delta function.
- The argument why the EDPA is accurate enough and under what condition should be presented. This is clearly not a general truth.
- Rephrase ``At the begin we''.
- Eq. (17), the channels are not defined.
- What about 3-body interactions?
- Sec. 3.1 `` three our separable'' --> ``three separable''.
- There. A ref. to the potentials is missing.
- The fitted AV18 NN interaction - in what energy range does the fitted interaction reproduce the data?
- End of 3.2 ASG--> AGS
- If Sec. 4 is included in the manuscript comparison with other published works must be included and discussed.

  • validity: good
  • significance: good
  • originality: ok
  • clarity: low
  • formatting: reasonable
  • grammar: below threshold

---

## Round 2 · Author Response

I am thankful to the referee for his/her report, but I cannot agree with all his/her remarks. The first claim is that I present preliminary results. But I would like to remind that this paper is a contribution to conference proceedings. As far as I know, it is quite normal to present preliminary results at conferences and publish them in the corresponding conference proceedings. When I "gain full confidence", I will publish a paper in an impact journal.

Below there are my replies to "few more minor comments": - In the introduction the author writes more accurate calculations are needed'', but no argument is given. Does she suspects there is an error in the reported results? does she has a better input? \I cite 3 other papers: one was performed using many-body G-matrix method for the three-body system, which is not well-grounded. In addition, AY potential, which was used there, does not describe antiK N experimental data. Another is a variational calculation, which evaluated only real part of the quasi-bound state using only real part of the (necessarily) complex antiK N potential, while the width of the state was somehow estimated after. The last one is a Faddeev-type AGS calculation, which should be the most accurate one. But the authors used two my antiK N potentials and obtained strongly different corresponding 4-body binding energies. Which I - keeping my experience with my own three-body Faddeev-type calculations using these potentials - cannot understand. That's why I am sure, that more accurate Faddeev-type calculation is necessary. This explanation is too long and detailed for the introduction of a conference paper, due to this I did not include it into the text of the paper.

  • separabelize'' is not a word. Same for its derivatives. \The sentences containing "separabelize" and its derivatives were reformulated.

  • The second term in Eq. (15) is just a delta function. \It is a delta function only at the fixed energy z=E_B, as is written above Eq.(14). Energy z in Eqs.(14,15) is arbitrary.

  • The argument why the EDPA is accurate enough and under what condition should be presented. This is clearly not a general truth. \Yes, it is not a general truth, that's why the sentence about accuracy of the EDPA is started by "According to the authors, ..." (the authors of the method, of cause, - I do not write about myself in such a way in the paper). The argument is probably their results obtained using the EDPA and compared with the exact ones.

  • Rephrase At the begin we''. \Done: "We started by writing down the system Eq.(\ref{4AGSsepVT}) for $18$ channels $\sigma_{\alpha}$ with $\alpha =$ $NN$ or $\bar{K}N$, considering three nucleons as nonindentical particles:"

  • Eq. (17), the channels are not defined. \The channels are written in an explicit form in Eq.(17). Does the referee insist on a special notation for every of them?

  • What about 3-body interactions? \"No three-body potentials were used since the four-body Faddeev-type equations are too complicated in their original form with "normal" pair potentials already." was added to the end of 3.1

  • Sec. 3.1 three our separable'' --> three separable''. \No: the potentials are ours since I had constructed them.

  • There. A ref. to the potentials is missing. \Thank you, added.

  • The fitted AV18 NN interaction - in what energy range does the fitted interaction reproduce the data? \"... at low energies up to $500$ MeV ..." was added into the sentence starting from "It reproduces Argonne v18"

  • End of 3.2 ASG--> AGS \Corrected

  • If Sec. 4 is included in the manuscript comparison with other published works must be included and discussed. \Here the referee contradicts himself/herself: how can I include comparisons with other results if I should (according to the referee) remove Section 4 completely - with all my preliminary results?.. On the other hand, comparison of preliminary results with others is of no use.

---

## Round 2 · List of Changes

The sentences containing "separabelize" and its derivatives were reformulated.

The sentence started "At the begin we" was rewritten:
"We started by writing down the system Eq.(\ref{4AGSsepVT}) for $18$ channels
$\sigma_{\alpha}$ with $\alpha =$ $NN$ or $\bar{K}N$, considering three nucleons
as nonindentical particles:"

"No three-body potentials were used since the four-body Faddeev-type equations
are too complicated in their original form with "normal" pair potentials already."
was added to the end of 3.1

References to the antiK N potentials were added to the 3rd sentence of 3.1.

"... at low energies up to $500$ MeV ..." was added into the sentence starting
from "It reproduces Argonne v18"

End of 3.2: ASG--> AGS was corrected

---

## Editorial Decision

published